# Differences in the Concentration of Anti-SARS-CoV-2 IgG Antibodies Post-COVID-19 Recovery or Post-Vaccination

**DOI:** 10.3390/cells10081952

**Published:** 2021-07-31

**Authors:** Andrzej Tretyn, Joanna Szczepanek, Monika Skorupa, Joanna Jarkiewicz-Tretyn, Dorota Sandomierz, Joanna Dejewska, Karolina Ciechanowska, Aleksander Jarkiewicz-Tretyn, Wojciech Koper, Krzysztof Pałgan

**Affiliations:** 1Faculty of Biological and Veterinary Sciences, Nicolaus Copernicus University, 87-100 Torun, Poland; prat@umk.pl (A.T.); monika_skorupa@umk.pl (M.S.); jdejewska@genetykatorun.pl (J.D.); 2Centre for Modern Interdisciplinary Technologies, Nicolaus Copernicus University, ul. Wilenska 4, 87-100 Torun, Poland; 3Non-Public Health Care Centre, Cancer Genetics Laboratory, 87-100 Torun, Poland; prezes@genetykatorun.pl (J.J.-T.); dorota.sandomierz@gmail.com (D.S.); karolinaciech@wp.pl (K.C.); aleksanderjt@gmail.com (A.J.-T.); 4Polish-Japanese Academy of Information Technology, 02-008 Warszawa, Poland; 5The Voivodeship Sanitary-Epidemiological Station in Bydgoszcz, 85-031 Bydgoszcz, Poland; wkoper@pwisbydgoszcz.pl; 6Department of Allergology, Clinical Immunology and Internal Diseases, Collegium Medicum, Nicolaus Copernicus University, 85-067 Bydgoszcz, Poland; palgank@cm.umk.pl

**Keywords:** COVID-19, Comirnaty, humoral response, cellular response, mRNA vaccine, spike protein

## Abstract

At the end of 2020, population-based vaccination programs with new generation mRNA-based vaccines began almost all over the world. The aim of the study was to evaluate the titer of anti-SARS-CoV-2 IgG antibodies against the S1 subunit of the virus’s spike protein as a marker of the humoral response in 477 patients and the concentration of interferon-gamma as an indicator of cellular response in 28 individuals. In our studies, we used serological enzyme-linked immunosorbent assays. IgG was measured in weeks 2 and 3 after the first dose and 1–5 weeks after the second dose of an mRNA vaccine in seropositive and seronegative individuals as well as in symptomatic and asymptomatic convalescents. High levels of antibodies were observed in 98% of our vaccinated cohort, and the presence of protective T cells was confirmed in the blood samples of all participants. The humoral immune response is diversified and is visible as early as 2–3 weeks after the first dose of the mRNA vaccine. The level of protection increased significantly after the second dose, with the increase being much greater in pre-vaccine healthy subjects and less in convalescents. In the second and third weeks after the second dose, the concentration of IgG antibodies was the highest, and in the following weeks, it decreased gradually. Regular serological measurements on eight subjects show that antibody titers are lower four months after vaccination than before the second dose.

## 1. Introduction

Severe acute respiratory syndrome coronavirus-2 (SARS-CoV-2), a member of the subgenus *Sarbecovirus*, causes coronavirus disease 2019 (COVID-19), and the first cases occurred in late 2019 [1]. On 11 March 2020, the World Health Organization (WHO) declared the SARS-CoV-2 outbreak to be a pandemic. As of 28 May 2021, almost 3,520,000 deaths and 169,480,000 cases of SARS-CoV-2 infection have been reported worldwide [2]. An understanding of the immune responses to SARS-CoV-2 and the coronavirus vaccines is necessary because of its rapid spread.

In response to SARS-CoV-2 infection, humans produce specific antibodies, CD4^+^ T cells, which activate high-affinity antibodies produced by B cells, and CD8^+^ T cells, which destroy infected cells [3,4,5]. SARS-CoV-2-specific antibodies are directed against the spike protein (S) and nucleocapsid (N). Special roles are played by neutralizing antibodies against the S1 subunit on the receptor-binding domain (RBD) that binds to angiotensin-converting enzyme 2 (ACE2) sites, thereby facilitating endocytosis, viral entry into host cells. After infection, antibodies can be detected in patients after 3 days when symptoms occur, and seroconversion in most of them appears within 7–14 days. In the acute phase of the disease, IgM antibodies develop and peak at 14 to 35 days and then begin to decline over the next 21 to 35 days. IgG antibodies peak at around 21 to 49 days after infection and, together with neutralizing antibodies, may persist for up to four months [6,7]. CD4^+^ T and CD8^+^ T cells specific for SARS-CoV-2 infections recognize peptides associated with the nucleocapsid, spike protein, and membrane proteins (M) of the virus and are present in most COVID-19 patients [6]. Specific CD4^+^ T cells differentiate into Th1 and Tfh cells: the former produce IFN-γ, and the latter, in addition to activating B cells, are crucial for the proper functioning of the neutralizing antibodies. Furthermore, CD4^+^ T cells help CD8^+^ T cells respond to infection, and a high concentration of specific infected cell-destroying CD8^+^ T cells gives a better prognosis for COVID-19 patients [4]. The determination of the required antibody titer and the duration of the immune system response, including the cellular response, are the basis for further research into better understanding the protective mechanisms, pathogenesis, and prognostic factors of COVID-19. This knowledge is also important for the development of effective treatments and vaccines [3,4,6,8].

During a global pandemic, mRNA vaccines are the fastest available vaccines due to their short production time and low biological requirements [1]. These mRNA-based vaccines avoid the risk of integrating viral genetic material into the host cell’s genome and are capable of producing pure viral protein. The technology of producing vaccines against COVID-19 in the form of lipid nanoparticles (LNP) enables the delivery of precise genetic information along with an adjuvant effect to antigen-presenting cells [1]. The SARS-CoV-2 vaccines are based on the virus’s mRNA, specifically on the fragment encoding the spike (S) protein, which attaches the virion to the host cell’s membrane [9]. Moreover, the S1 subunit of the S protein contains an immunologically relevant receptor-binding domain (RBD), which is a key antibody target [1]. According to clinical studies, subjects developed a strong dose-dependent antibody response to the S protein after the first and second inoculations [10]. Neutralizing antibodies were found in all subjects after the second inoculation, and the antibody titers were equal to or greater than the neutralizing antibody titers of COVID-19 patients [10].

The mRNA vaccine is a Comirnaty concentrate from Pfizer and BioNTech. One dose (0.3 mL) contained 30 micrograms of the COVID-19 mRNA vaccine. The active substance of the preparation is the mRNA that encodes the spike protein of the virus and acts as an antigen [11,12]. The vaccine also contains four types of fats in the form of lipid nanoparticles: (4-hydroxybutyl)azanediyl, bis(hexane-6,1-diyl), bis(hexyl-2-decanoate), (ALC-0315),2-((polyethylene glycol)-2000)-N, N-ditetradecylacetamide (ALC-0159), 1,2-distearoyl-sn-glycero-3-phosphocholine (DSPC), cholesterol and other substances such as potassium chloride, potassium dihydrogen phosphate, sodium chloride, disodium phosphate dehydrate, saccharose, and water for injections [11]. Comirnaty is indicated for the active immunization of a person from the age of 16 years for the prevention of COVID-19 disease caused by the SARS-CoV-2 virus. The product is administered intramuscularly into the deltoid muscle after dilution, and a second dose is administered after at least 21 days [11,12].

The immune response to COVID-19 is poorly understood. There is insufficient knowledge about safe and immunologically effective vaccination strategies against SARS-CoV-2, and it is not known which vaccination strategies will be most effective [13]. Moreover, there is still insufficient information on the short- and long-term effects of these mRNA vaccinations. The aim of the research was to determine the humoral and cellular responses in vaccinated persons and in convalescents.

## 2. Materials and Methods

### 2.1. Human Subjects

The research group consisted of 477 adult volunteers, 362 women and 115 men. All were assessed for medical decision-making capacity using a standardized, approved assessment and voluntarily provided informed consent prior to being enrolled in the study. The patients represented research subgroups as shown in Table 1: 24 healthy and unvaccinated individuals; 15 persons with confirmed SARS-CoV-2 infection but no symptoms (unvaccinated); 82 persons who had COVID-19 and had multiple symptoms of infection (unvaccinated); 19 persons who had been vaccinated within 2 or 3 weeks but did not have COVID-19; 203 persons who had the second dose within 1 to 5 weeks before blood sampling for analysis and did not have COVID-19; 124 patients who were confirmed to be infected with SARS-CoV-2 and who were vaccinated at corresponding time points.

### 2.2. Quantitative Determination of IgG Antibodies against SARS-CoV-2

The starting material for analyses was serum obtained after centrifuging whole blood in clot tubes for 5 min at 4000 rpm. The serum was then carefully removed from the cell pellet and used. Depending on the patient subgroup, the sera were diluted 5-, 10-, 20-, 50- or 100-fold. An enzyme-linked immunosorbent assay (ELISA) was used to quantify the in vitro quantification of human IgG antibodies to SARS-CoV-2. The tests were performed using the commercial automated analyzers EUROIMMUN Analyzer I-2P and the Anti-SARS-CoV-2 QuantiVac ELISA kit (IgG) EUROIMMUN (Lübeck, Germany) according to the manufacturer’s instructions. The test reaction wells were coated with the S1 domain of the SARS-CoV-2 spike protein recombinantly expressed in the human cell line HEK 293 and enables specific detection of IgG antibodies against SARS-CoV-2 using the S1 domain of the spike protein, including the immunologically relevant receptor-binding domain (RBD), which represents important target antigen for virus-neutralizing antibodies. In the first step, reaction wells were incubated with diluted patient samples (1:101 dilutions) for 60 min at 37 °C. In the presence of IgG antibodies bound to the antigens on the surface of the well. After washing with a solution (3 × 450 µL), the bound antibodies were incubated for 30 min at 37 °C with 100 µL of an anti-human IgG antibody-peroxidase conjugate. After subsequent washing (3 × 450 µL), 100 µL of substrate/chromogen solution was added to each well and incubated for 30 min at room temperature (RT). During the last stage of the procedure, 100 µL of stopping solution was added to each well. Photometric determination of the color intensity was performed at a wavelength of 450 nm. The results were expressed in binding antibody units (BAU)/mL. A six-point standard curve (3.2–384 BAU/mL) was performed in parallel, and positive and negative control in the form of human IgG was used. 

### 2.3. Cellular Response Analysis

The EUROIMMUN SARS-CoV-2 Interferon-Gamma Release Assay (SARS-CoV-2 IGRA) kit was used to assess the cellular immune response against SARS-CoV-2. The IGRA test was used to determine the activity of pathogen-responsive T cells by detecting interferon-gamma (IFN-γ). The research material was freshly drawn whole blood. T cells capable of recognizing antigens of this pathogen were present in the blood of a patient who had come in contact with the SARS-CoV-2 virus. This process is associated with the synthesis and release of IFN-γ. The detection and quantification of IFN-γ secreted after in vitro stimulation of T cells was the essence of this test. The determination was divided into two stages: in vitro stimulation of T lymphocytes with a SARS-CoV-2-specific antigen (the S1 subunit of the SARS-CoV-2 protein) and measurement of secreted IFN-γ by T cells using ELISA.

#### Cell Stimulation

The starting material for analyses was fresh whole blood collected in lithium heparin tubes. SARS-CoV-2 IGRA stimulation tube EUROIMMUN (Lübeck, Germany) was used to stimulate the cells, according to the manufacturer’s instructions. The kit contained three test tubes for one sample. The first (BLANK) did not contain any activating ingredients for immune cells. The plasma obtained from it was used to determine the individual interferon-gamma background. The second (TUBE) was coated with the components of the S1 domain. If activatable lymphocytes were present in the blood, they would have been stimulated to secrete interferon-gamma during incubation. The third (STIM) was coated with a mitogen to induce non-specific interferon-gamma secretion. The plasma obtained from this tube was used to verify whether the sample contained a sufficient number of cells and whether they had a sufficient ability to become active. Then, 500 µL of whole blood was pipetted into the three tubes and incubated at 37 °C for 24 h. After this time, the tubes were centrifuged for 10 min at 12,000 rpm. The plasma interferon-gamma concentration from the BLANK tube represented the individual’s interferon-gamma background and therefore had to be subtracted individually from the plasma concentration obtained in the TUBE and STIM tubes. After subtraction, the interferon-gamma concentration in the STIM tube had to be significantly higher than the BLANK value alone to consider the immune cell count and stimulation in the whole blood sample to be sufficient.

### 2.4. Interferon-Gamma ELISA

The level of interferon-gamma was determined by an ELISA test performed on a commercial automatic EUROIMMUN Analyzer I-2P with the use of the Interferon-gamma ELISA kit, according to the manufacturer’s instructions. The test kit contains a microplate with reaction wells coated with anti-interferon-gamma monoclonal antibodies. In the first reaction step, calibrators, controls, and plasma samples diluted in a sample buffer (1:5) were added to the coated reaction wells to bind interferon-gamma and were incubated for 120 min at RT. Then, the wells were washed with a washing buffer (5 × 450 µL). In the next step, 100 μL of biotin-labeled anti-interferon-gamma antibodies were added and incubated again for 30 min at RT. The washing was repeated (5 × 450 µL), then 100 µL of streptavidin-HRP was added and incubated for 20 min at RT. Photometric determination of the color intensity was carried out at a wavelength of 450 nm. The color intensity was proportional to the interferon-gamma concentration. The results were expressed in mIU/mL.

### 2.5. Statistical Analysis

One measurement was performed for each patient and based on the medical questionnaire. The patient was assigned to a group according to vaccination and COVID-19 status. The mean, median, minimum and maximum values were determined for each group. To determine the significance of differences between experimental subgroups, a one-way analysis of variance (ANOVA, *p* < 0.05) was performed. Spearman’s test (two-tailed) was used to determine the correlation. Statistical analyses were performed using the IBM SPSS Statistics software (version 27.0.1.0)

## 3. Results

### 3.1. Characteristic of Study Group

Serum samples were collected from 477 individuals between 2 February and 9 March 2021 in Toruń, Poland. The main participants were medical professionals, employees and patients of nursing homes, sanitary and epidemiological inspectors, and volunteers. People taking part in the study completed a questionnaire, which included information on: age, date of COVID-19 onset and symptoms of infection, date of administration of vaccine doses, and coexistence of chronic diseases. The age of the participants ranged from 18 to 93 years, and samples were taken at different times after confirming SARS-CoV-2 infection or mRNA vaccination: 18.2% of participants 35 years or younger, 25.4% were 36–45, 22.2% were 46–55, 23.1% were 56–64, and 11.1% were 65 and older. In all, 362 women (75.9%) and 115 men (24.1%) participated. The study group included symptomatic and asymptomatic recoveries and persons vaccinated with the first and second dose of mRNA vaccines. Of the 477 serum samples, 356 came from those who had been vaccinated between 2 weeks after the first dose and 5 weeks after the second dose, 134 were from those who had recovered, and 121 were obtained from unvaccinated individuals. Healthy people were chosen as the control group. The detailed structure of the study group, broken down into subcategories by age and sex, is presented in Table 1. The group included 16 people who became ill with COVID-19 within 2 weeks of taking the first dose, 1 who became ill 4 weeks after taking the first dose, and 2 people who tested positive at 4 and 6 weeks after taking the second dose of mRNA vaccines. These people fell ill a few days after the serological test, which showed that they had high antibody titers 262.2 BAU/mL and 1106 BAU/mL, respectively).

### 3.2. Comparisons of an Antibody Level between Subgroups

Anti-SARS-CoV-2 IgG antibody concentrations ranging from 0 to 38,400 BAU/mL were analyzed in the study (Figure 1, Table 2). Concentrations below 25.6 BAU/mL (negative result) were found in people who were not vaccinated and did not suffer from SARS-CoV-2 infection, 12 who recovered (infection confirmed in October and November 2020), as well as in 3 people who had only taken the first dose of mRNA vaccine. Relatively low primary humoral immunity was found in three patients 2 weeks after taking the second dose (78 BAU/mL for an 86-year-old woman, 89 BAU/mL for an 80-year-old woman, and 106.02 BAU/mL for a 46-year-old man, respectively).

In seronegative subjects, in the third week after immunization, the mean level of antibodies was higher than in seropositive subjects without vaccination, which confirms the effectiveness of the vaccines in inducing a humoral response. From 10 to 14 days after the second dose, a 10-fold increase in IgG level was obtained.

The highest levels of anti-SARS-CoV-2 IgG were found in vaccinated subjects who had undergone SARS-CoV-2 infection, both after the first and second doses, regardless of the week of vaccination (Figure 1, Table 2). Two weeks after the first dose, the median level of antibodies in seronegative subjects was lower than in seropositive subjects without vaccination. In the third week after the first dose, vaccinated convalescents had a titer of IgG antibodies more than 11-fold higher than in those who had not received the first dose in the same period (348.00 BAU/mL vs. 3989.00 BAU/mL; *p* = 0.008). After the first dose, the median as well as the mean (See Appendix A) individual seropositive anti-SARS-CoV-2 IgG concentrations each week were higher than for seronegative subjects after the second dose of the mRNA vaccine (Figure 1).

### 3.3. Correlation of the Antibody Titer with Age and Sex

The correlation between the concentration of IgG antibodies and the age and sex of the participants was analyzed. There was no significant correlation (*p* > 0.05) of the titer of antibodies against the S protein, although a lower concentration of antibodies of this class was noticeable in men compared to women (Figure 2; Appendix A) in each of the analyzed subgroups, but the disproportionate sizes had to be taken into account. Similarly, for each of the compared categories, no significant correlation was found between age and the concentration of anti-SARS-CoV-2 IgG (*p* > 0.05). Nevertheless, a noticeable trend was the highest concentration values for people aged 36–45 and 46–55. (Figure 2). For people over 65, slightly lower antibody titers were found, but the difference was noticeable after the age of 80 and in those with chronic diseases, especially diabetes, thyroid disease, and ulcerative enteritis (data not shown).

### 3.4. Monitoring of the Humoral Response in the First Weeks after Vaccination

For the eight vaccinated persons (COVID laboratory employees) who had not been infected with SARS-CoV-2, weekly and then monthly determinations of IgG antibody levels for the first 2 months after vaccination were performed to understand the dynamics of immunization. All of them showed a several-fold increase in the level of anti-SARS-COV-2 IgG antibodies compared to the previous measurement until the second week after receiving the second dose, when the greatest changes (5–10-fold increase) in IgG concentration were noted in the first and second weeks afterward. Between the second and third weeks after the second dose of the vaccine (6 weeks after the first dose of vaccine), all participants had a significant decrease in anti-SARS-CoV-2 antibody titers (Figure 3). In the third week after the first dose, the mean concentration was 998.89 BAU/mL (range 83.83–2845 BAU/mL). In the second week after the second dose, it was highest with a mean of 6056.18 BAU/mL (range 1889–12,650.50 BAU/mL). After 10 weeks of vaccination, antibody levels had dropped to a mean level of 1758.66 BAU/mL (range 320.00–3840.00 BAU/mL). After 4 months, the concentration of antibodies in all participants of the study fell below the level observed before the administration of the second dose.

### 3.5. Humoral Immunity and Cellular Immunity

Cellular immunity analysis was performed for selected patients. Th-cell activity was analyzed indirectly by measuring the concentration of interferon-gamma secreted by activated lymphocytes after 24 h of in vitro stimulation. The mean concentration of IFN-γ in the non-antigen stimulated samples was 18.17 mIU/mL (range 0.50–89.08 mIU/mL). After stimulation by the S1 antigen, the concentration of interferon-gamma in the samples increased significantly (*p* < 0.001). The mean concentration was 1625.00 mIU/mL (range 11.75–2499.25 mIU/mL). In convalescents, the mean level of IFN-ɣ was 1210.53 (range 91.56–2498.53 mIU/mL), and there was a correlation between the determined amount of gamma interferon and the time after the onset of COVID-19. The lowest concentrations were obtained for the sick in October and November 2020. In people who received two doses of the mRNA vaccine, the mean concentration of IFN-ɣ was similar to the level described in convalescents at 1172.73 mIU/mL (range 11.75–2485.92 mIU/mL). The highest concentration of IFN-ɣ was 1854.52 mIU/mL (range 168.41–2499.25 mIU/mL) detected in vaccinated SARS-recovered. Despite the high titer of antibodies and the concentration of interferon-gamma (Figure 4), three people in this group became infected with the SARS-CoV-2 virus. One of these patients was reinfected with the virus within 6 months of recovering from COVID-19 and 4 weeks after receiving the vaccine, which confirmed the lack of sufficient immune protection.

## 4. Discussion

Vaccination programs are implemented at different rates. By the end of May, 1.9 billion doses had been administered worldwide (19.9 million in Poland), and 426 million people had been vaccinated with the full dose (6.87 million in Poland). The global percentage of the population after full vaccination was 5.5% (18.1% in Poland) [14]. Vaccines against COVID-19 produced by Pfizer/BioNTech use mRNA technology. According to the characteristics of the drug, the minimum time needed to obtain protection after the second dose for the Comirnaty vaccine is 7 days. The onset of protection was observed approximately 14 days after vaccination. Clinical trials showed almost 95% effectiveness in preventing severe COVID-19 disease in people without prior infection. After the first dose, the effectiveness of the preparation was estimated at approximately 52% [15]. How long immunity is induced by SARS-CoV-2 infection remains unclear at this stage, but antibodies are expected to last for at least six months (as in the case of a COVID-19) to potentially several years. There is also insufficient information on protection against the emerging new variants of the virus.

It is known from studies on animal models and observations of convalescents that the most important therapy, especially in the vaccination strategy, are neutralizing IgG antibodies directed against the spike (S) protein that can block infectivity) [15,16,17] and SARS-CoV-2-specific T cells respond to many viral proteins (especially dominant S-protein-specific CD4 + T cells) [4,18]. The subpopulation of these lymphocytes is additionally correlated with the humoral response manifested by the presence of anti-SARS-CoV-2 antibodies [19]. Petrone et al. [20] found a correlation between the B-cell and T-cell responses to SARS-CoV-2 antigens; thus, analogous to similar viral infections, they confirmed the strict relationship between the two immune compartments in COVID-19. In this study, we analyzed the elements of the immune response primarily in vaccinated (seropositive and seronegative) individuals and compared them with determinants of immunity in convalescents (vaccinated and unvaccinated). The basic determinant of immunization as a result of disease or vaccination in our study was the analysis of the humoral response expressed by the concentration of IgG antibodies against the S protein. The purpose of this analysis was to screen the primary immune response and thus provide a cross-sectional analysis of the dynamics of B-cell responses in each of the 16 compared patients groups. In the second stage, the activation of T lymphocytes was analyzed in selected people (in three representative groups). Based on the results, a high degree of heterogeneity of immune responses was found. After vaccination, the parameters of the humoral response were measurable in all participants, which confirmed the effectiveness of the mRNA vaccine in activating B lymphocytes to produce antibodies and T lymphocytes to secrete gamma interferon. Additional assessment of the cellular immune response (detection of interferon-gamma, including the determination of pathogen-responsive T-cell activity) confirmed post-vaccination and post-COVID-19 immunization at the cellular level in all subjects. The response was variable, but we did not observe such wide differences as in the case of anti-SARS-CoV-2 antibodies.

Primary humoral immunity, one of the indicators of which is the presence of IgG antibodies, appeared 2 weeks after receiving the mRNA vaccine in 66.8% of people vaccinated with the first dose of Comirnaty and after 3 weeks in all 11 people. Researchers working on the clinical trials for the Comirnaty vaccine observed vaccine effectiveness of 52% between the 21 days of the first and second doses. Based on independent U.K. studies, it is estimated that the Pfizer/BioNTech vaccine may be more effective after the first dose than previously thought. In this study, the effectiveness of the first dose of the vaccine 15 days after receiving it was actually closer to 89 to 91 percent [21]. Researchers at the University of Sheffield and University of Oxford, in cooperation with the U.K. Coronavirus Immunology Consortium (U.K.-CIC), tentatively concluded, based on observational studies conducted in the U.K. in which healthcare workers were vaccinated against COVID-19, that the first dose of the vaccine may provide immune protection against a severe course of COVID-19. The study was conducted on a group of 237 people, some of whom had previously been infected with SARS-CoV-2 and some who had never suffered from COVID-19. The above-mentioned researchers, similar to our observations, obtained the strongest immune response in those who had been infected with SARS-CoV-2 before vaccination. After one dose of the Pfizer/BioNTech vaccine, the levels of T cells clearly increased compared to the levels seen in the blood of people who had been vaccinated but were uninfected [22].

Based on the concentration of anti-SARS-CoV-2 antibodies, it was found that patients who experienced symptomatic SARS-CoV-2 infection, both after receiving the first and second doses, regardless of the week of vaccination, had significantly higher antibody titers compared to seronegative people. Our observations are consistent with the studies of Angyal et al. [22], who published data showing that in people who were vaccinated after contracting COVID-19, antibody responses after the first dose of Pfizer/BioNTech vaccine were 6.8 times higher, and T-cell responses were 5.9 times higher than in people who had never had the disease. In contrast, among those who did not become sick but received a single dose of mRNA vaccine, the level of protection was similar or higher than that observed after natural infection. These researchers also did not find any correlation between age and the intensity of the humoral or cellular response. Our observations are also consistent with the results published by Krammer et al. [23], Saadat et al. [24], Ebinger et al. [25], Mazzoni et al. [26], and Gobbi et al. [27]. Both our study and the available studies show that the titer of antibodies in seropositive people after the first dose of the vaccine is about 10 times higher than that in vaccinated people who had never had the disease. Based on these results, one can assume that a prior SARS-CoV-2 infection triggered the immune system to a very strong response to a single dose of the vaccine. The first dose, given in to people whose immune systems had already been stimulated by the natural infection, had a similar effect when given as a second “booster” dose. Moreover, administration of the second dose in seropositive persons did not significantly increase antibody concentration. Confirmation of these observations could constitute a premise for the optimization of the vaccination program, in which decisions about taking vaccine doses should be based on the analysis of primary indicators of immunity. Another important aspect pointed out by Krammer et al. [23] is that taking the first dose of an mRNA vaccine by seropositive people could protect people who formerly suffered from COVID-19 from the negative effects of taking the second dose of the preparation. It appears that the added benefit of delaying or eliminating the second dose in highly immunized individuals would also be to increase the distribution of vaccine stocks among multiple individuals. Nevertheless, this approach requires further research, including the analysis of factors influencing overall immunity or vaccine efficacy. Current FDA recommendations recommend adherence to a dosing schedule that has been tested in clinical trials [28]. On the other hand, we are seeing a significant decline in antibody levels in pre-vaccination seronegative individuals with regular serological monitoring. A third “booster” dose appears to be inevitable for these people in the coming months. Interestingly, in another study (unpublished data), we observed that the decrease in antibody levels of seropositive people before vaccination was not that intense, which suggests a greater persistence of the post-vaccination response in triple-immunized people (as a result of natural disease and the intake of two doses of mRNA vaccine). In view of the above, it seems that full vaccinations in convalescents are justified, and vaccinating people without previous COVID-19 disease with a third dose is probably necessary. Doria-Rose et al. [29] based on interim results from a phase 3 trial of the Moderna mRNA-1273 calculated the half-life of vaccine-binding antibodies and determined the lifetime of the vaccine immune response of 6 months after second dose.

The group of SARS-recovered patients included both those with high serum levels of antibodies and those whose IgG titer may suggest a loss of immunity acquired after COVID-19 and thus indicate the need for vaccination. Antibody levels below <35.2 BAU/mL (negative or uncertain result) were detected in symptomatic convalescents (10 patients) who had been ill 5–6 months prior to serological examination. This observation is in accordance with the reports contained in ECDC Technical Report [30]. The currently available results of cohort studies confirm that the protective effect of natural SARS-CoV-2 infection ranges from 81% to 100%, begins on day 14 after infection, and lasts for a period of five to eight months [5,30]. Unfortunately, relatively low titers of IgG antibodies were also determined in five asymptomatic survivors after a period of several weeks after the positive test results, which may indicate a high risk of viral reinfection. People who have had COVID-19 should be vaccinated to ensure long-term and strong immunity. Chia et al. [31] noted that in convalescents groups, it is possible to distinguish five different patterns of the dynamics of neutralizing antibodies, and their modeling may influence the prediction of individual immunity longevity in convalescents, and thus the decision to vaccinate this group of patients. The persistence of neutralizing antibodies in SARS-recovered people was related to the severity of the disease, which we also saw in our study) and to the sustained levels of proinflammatory cytokines, chemokines, and growth factors. Chia et al. [31] also observed that, despite the different dynamics of neutralizing antibodies in the different groups, T-cell responses were similar. Therefore, it seems likely that analogous dynamics of the humoral and cellular responses may also apply to the post-vaccination immune response.

Our results showed that the humoral immune response induced by natural infection in convalescents was significantly enhanced by a single dose, and vaccination significantly improved immune cell responses after infection. Moreover, studies by Angyal et al. [22] showed that the first administration of the vaccine strengthens the cellular response, as well as in vitro, it strengthens the neutralizing properties in relation to the variant B.1.351 (South African). This aspect is pointed out by Skelly et al. [32], who tested the neutralization strength of antibodies resulting from natural SARS-CoV-2 infection and immunization with the Pfizer/BioNTech vaccine. They noted that there is a difference in the humoral (decreased neuralization) and, to a lesser extent, cellular responses to variants of the B1.1.7 (U.K.) and B1.351 lines. The authors attributed these differences to the strength of the homotypic antibody responses. Thus, it is speculated that the new SARS-CoV-2 variants may avoid the protective neutralizing responses resulting from natural infection, and to a lesser extent, immunization. Hence, as the authors emphasize, there is a need to induce a vaccine immune response. Interestingly, in vaccinated convalescents, even after the first dose, neutralizing antibodies were shown to be more effective against B.1.1.7 and B.1.351 than in those who did not have COVID-19 [33].

In the serological test, three had significantly lower spike-specific IgG levels compared to other vaccinated persons (concentration: 78–106.02 BAU/mL). Based on these results, these people did not acquire significant humoral immunity and may still be at risk of infection despite vaccination. Hence the need to conduct special serological surveillance in people aged 65+ or with coexisting chronic diseases, and perhaps to consider the need to take further doses to ensure protective properties. Blain et al. [34] focused on a group of nursing home residents and confirmed that one dose of an mRNA vaccine may also be sufficient. At the same time, they pointed out that testing the level of antibodies with neutralizing properties should be carried out for these patients, especially for those with an unknown history of infection. Monin et al. [35] emphasized the importance of prioritizing cancer patients for an early (21-day) second dose due to the low efficacy of the single dose in this group. Worrying is also the fact of confirmed COVID-19 cases for three participants despite vaccination, high rates of cellular and humoral responses. However, no variant of the virus has been identified that overcame the immune response mechanisms in these individuals. The course of COVID-19 in these people was mildly symptomatic, and no one required hospitalization. SARS-CoV-2 infections after vaccination have been reported sporadically [36,37], but they raise important issues regarding the duration of immunity after natural infection and the extent of protection after vaccination, as well as the transmission of the virus. We conducted regular immunological surveillance for eight seronegative participants and observed that 4 months after the first vaccinations, anti-SARS-CoV-2 IgG levels were lower and continued to decline. In Poland, over 80,000 vaccinated people (0.9%) tested positive for SARS-CoV-2. However, more than half of these infections occurred up to 14 days after the first dose. According to data from the Ministry of Health, no deaths due to COVID-19 among vaccinated people were recorded in Poland until May 18. In a cohort of healthcare professionals regularly tested by PCR, the absolute risk of a positive SARS-CoV-2 test after vaccination was 0.97%–1.19% and, as Keehner et al. [38] emphasized, these rates are higher than the risk reported in the vaccine studies mRNA-12731 and the BNT162b vaccine, which may be a consequence of the test frequency. A similar COVID-19 incidence rate was reported among nursing home residents. According to data provided by White et al. [39], 4.5% of cases were reported within 0 to 14 days of the first dose and 1.4% of cases within 15 to 28 days. Among the subjects vaccinated with both doses of the vaccine, 1.0% were PCR-positive within 0 to 14 days after the second dose, and 0.3% had a “breakthrough” infection 14 days after full vaccination.

## 5. Conclusions

Testing the concentration of antibodies to the S protein in both convalescents and vaccinated patients enabled the analysis of the course of the humoral immune response to COVID-19. Quantitative testing of anti-SARS-CoV-2 IgG antibodies determines whether the patient responded to the vaccination, and if so, how intensely. It also enabled the assessment of the humoral immunity acquired after undergoing SARS-CoV-2 infection. By testing anti-SARS-CoV-2 antibodies, it was possible to determine the concentration of that provided protection against infection, as well as to make rational decisions about booster doses of the vaccines. The greatest benefit of the research is the ability to quantify the acquisition of humoral immunity to SARS-CoV-2 as a result of infection or vaccination. Antibody testing is not required in the context of vaccination, but knowledge of the immune status before and several weeks after the last dose may nevertheless allow inferences about the immune response to immunization and provide an indication of the degree of immunity. Due to the lack of data on the persistence of immunity acquired as a result of a vaccine reaction, it is also important to monitor the level of antibodies over time (especially among healthcare professionals, people over 65 years, and the chronically ill), as a booster dose may be needed. The results of the research could be useful for developing new vaccination recommendations and the need to continue them on a larger scale. The diversity of immune responses shows the need for research, the inherent element of which will be immunological monitoring of the durability of disease resistance or protection against its severe course in vaccinated people or susceptibility to reinfection in COVID-19 convalescents. By analyzing the level of antibodies, it was possible to identify people who are already immunized as well as those who had not acquired immunity as a result of vaccination, and those who may have lost the acquired immunity after contacting SARS-CoV-2. In our opinion, such a test should be an integral part of the assessment of immunological parameters, especially before making an informed decision about vaccination or its delay in convalescents, as well as the assessment of the durability of immune protection.

## Figures and Tables

**Figure 1 cells-10-01952-f001:**
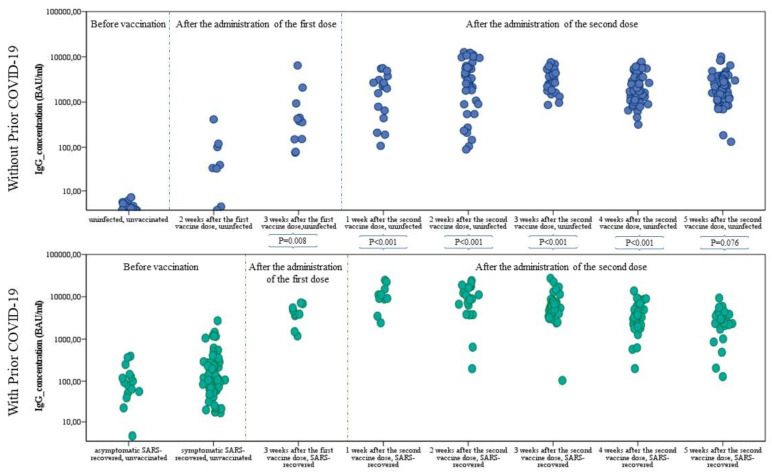
Comparisons of antibody levels between the analyzed patients’ subgroups. Blue indicates the concentration of anti-SARS-CoV-2 IgG antibodies in seronegative individuals before vaccination. Individuals with confirmed SARS-CoV-2 infection prior to receiving the mRNA vaccine dose, as well as for seropositive unvaccinated individuals, are marked in green.

**Figure 2 cells-10-01952-f002:**
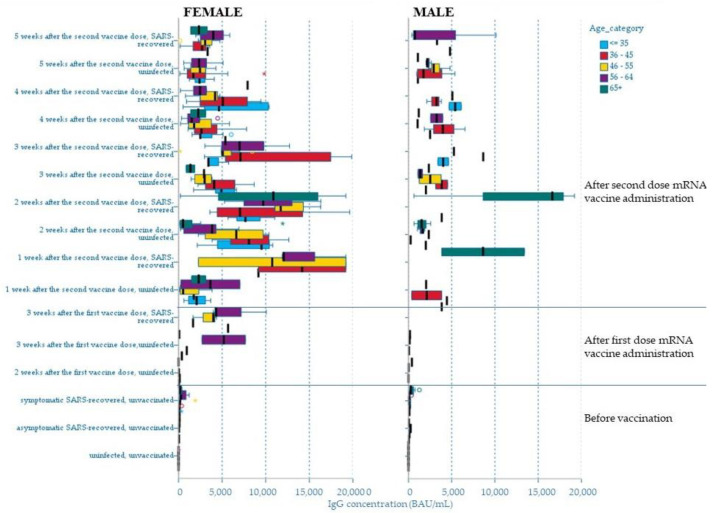
Distribution of the mean values of anti-SARS-CoV-2 IgG antibodies in relation to age and the distribution of anti-SARS-CoV-2 IgG concentrations by sex of the study participants.

**Figure 3 cells-10-01952-f003:**
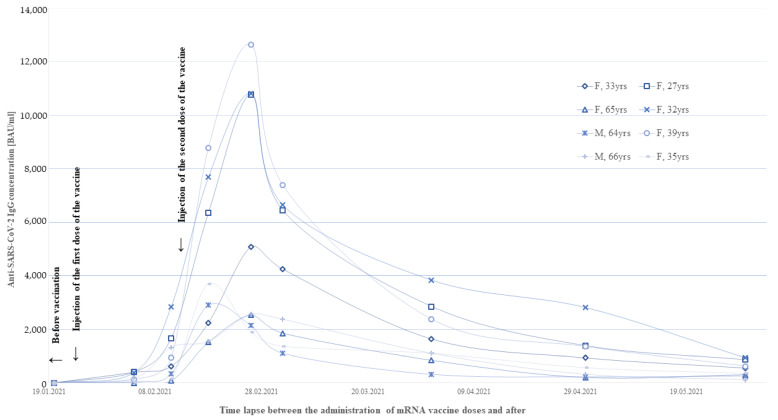
IgG-anti-SARS-Cov-2 concentrations in the first 4 months after receiving the mRNA vaccine for 8 seronegative individuals.

**Figure 4 cells-10-01952-f004:**
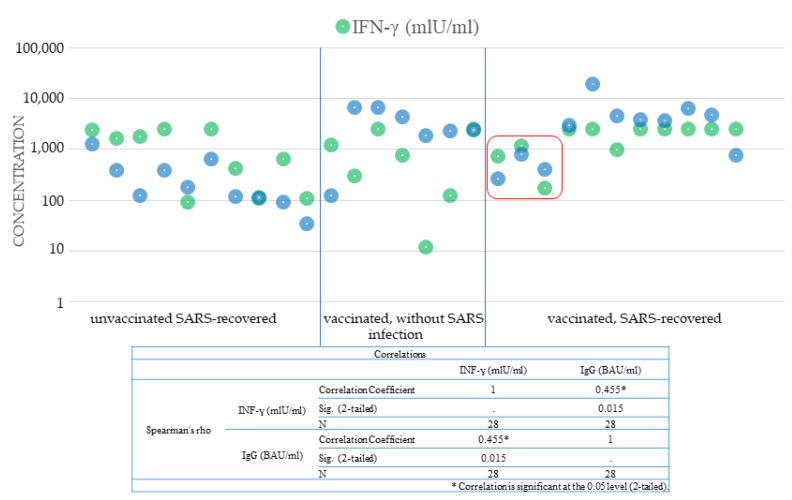
Comparison of anti-SARS-CoV-2 IgG antibody titers against S1 protein and interferon-gamma concentrations released after stimulation of Th lymphocytes in 28 individuals (patients who recovered from SARS. vaccinated persons without previous viral infection. and vaccinated SARS-recovered). The red frame marks 3 people who were infected with the SARS-CoV-2 virus despite vaccination.

**Table 1 cells-10-01952-t001:** Characteristics of the frequency of the studied subgroups, taking into account the sex and age of the patients.

		Female					Male				
		Age (Years)									
SARS-CoV-2 and Vaccination Status	N	≤35	36–45	46–55	56–65	65+	≤35	36–45	46–55	56–65	65+
Unvaccinated											
Without prior SARS-CoV-2 infection	24	4	2	3	5	0	2	4	2	1	1
Convalescents	97	13	14	14	16	7	4	7	8	7	7
Vaccinated with the first dose (2–3 weeks after vaccination)											
Without prior SARS-CoV-2 infection	19	6	2	1	2	2	0	2	0	2	2
Convalescents	9	1	1	3	3	0	0	0	0	0	1
Vaccinated with the second dose (1–5 weeks after vaccination)											
Without prior SARS-CoV-2 infection	203	32	38	40	36	13	5	12	9	11	7
Convalescents	125	14	35	23	24	8	6	4	3	3	5
Total	477	70	92	84	86	30	17	29	22	24	23

**Table 2 cells-10-01952-t002:** Median, maximum, and minimum concentrations of anti-SARS-CoV-2 IgG antibodies in the compared subgroups.

**COVID-19 and/or Vaccination Status**	**IgG_concentration (BAU/mL)**				
	**Mean**	**Median**	**Minimum**	**Maximum**	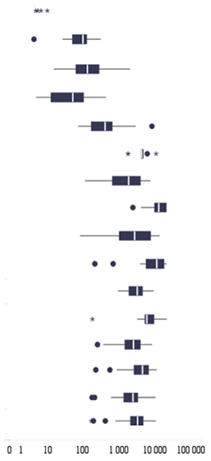
uninfected, unvaccinated	3.57	3.20	3.20	8.56	
asymptomatic SARS-recovered. unvaccinated	112.59	94.95	3.20	296.40	
symptomatic SARS-recovered. unvaccinated	236.52	124.33	13.63	1,920.00	
2 weeks after the first vaccine dose. uninfected	91.36	46.99	3.55	404.45	
3 weeks after the first vaccine dose.uninfected	1,209.40	384.00	67.62	7,680.00	
3 weeks after the first vaccine dose. SARS-recovered	4,397.00	3,989.00	1,666.00	10,068.00	
1 week after the second vaccine dose. uninfected	2,228.53	1,747.20	106.02	7,045.00	
1 week after the second vaccine dose. SARS-recovered	11,940.33	11,973.25	2,280.30	19,200.00	
2 weeks after the second vaccine dose. uninfected	4,261.08	2,563.20	78.00	12,650.50	
2 weeks after the second vaccine dose. SARS-recovered	10,241.42	10,375.75	198.70	19,638.00	
3 weeks after the second vaccine dose. uninfected	3,327.63	3,039.60	880.00	8,677.00	
3 weeks after the second vaccine dose. SARS-recovered	7,442.47	5,516.00	168.50	19,865.00	
4 weeks after the second vaccine dose. uninfected	2,658.50	2,379.50	235.50	7,805.00	
4 weeks after the second vaccine dose. SARS-recovered	4,490.81	4,196.50	210.50	10,391.50	
5 weeks after the second vaccine dose. uninfected	2,474.46	2,294.50	166.00	9,836.50	
5 weeks after the second vaccine dose. SARS-recovered	3,190.51	3,096.25	160.00	10,135.50	

## Data Availability

The data presented in this study are available on request from the corresponding author. The data are not publicly available due to their containing information that could compromise the privacy of research participants.

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
