# Peer review of "Differences in the Concentration of Anti-SARS-CoV-2 IgG Antibodies Post-COVID-19 Recovery or Post-Vaccination"

_cells, 2021, doi:10.3390/cells10081952_

Round 1

Reviewer 1 Report

The paper by Andrzej Tretyn and coworkers reports an interesting study on immunization post-COVID or/and post-vaccination analyzing the concentration of anti-SARS-CoV-2 IgG 2 antibodies in a cohort of 477 patients. The aim of the research was to determine the humoral and cellular responses in vaccinated persons and in convalescents. The analysis is quite interesting and the results sound.

Moreover, since there is still insufficient information on the short- and long-term effects of vaccination, the valuation of long term immunity acquired with the use of the mRNA vaccine developed by Pfizer, is of primary importance to help making decisions about the necessity of administrating further doses (third only??)

Therefore I think that the ms can be published as it is

.

Author Response

Dear Editor and Reviewers,

we are thankful for your evaluation and for your comments.

We were thrilled to learn that it has been conditionally accepted for revision and appreciate the helpful comments and suggestions provided by the reviewer and your editorial team. We have carefully reviewed all the comments and suggestions and revised our manuscript accordingly. Below you will find detailed replies to each reviewer comment and the appropriate reference to the changes made within the manuscript itself. In summary;

  • we have prepared new figures and tables, which in our opinion should be more legible;
  • we have updated the discussion and reference list to reflect the latest publications in the field of post-vaccination serology tests;
  • we changed the description of the methodology of gamma interferon assays, so that it was clear that lymphocyte stimulation was performed on the fresh whole blood sample, and after 24 hours of incubation, we obtained plasma in which we measured the concentration of gamma interferon;
  • we made a linguistic correction of the text.

We believe that the revised version of our manuscript was improved and a detailed response to each of the reviewers' comments will dispel any doubts. We hope that the manuscript is now acceptable for revision and we would like to thank you for your time spent in considering our manuscript.

Sincerely,

Joanna Szczepanek and Andrzej Tretyn,

on behalf of all co-authors

Responses for the Reviewer 1:

The paper by Andrzej Tretyn and coworkers reports an interesting study on immunization post-COVID or/and post-vaccination analyzing the concentration of anti-SARS-CoV-2 IgG 2 antibodies in a cohort of 477 patients. The aim of the research was to determine the humoral and cellular responses in vaccinated persons and in convalescents. The analysis is quite interesting and the results sound.

Moreover, since there is still insufficient information on the short- and long-term effects of vaccination, the valuation of long term immunity acquired with the use of the mRNA vaccine developed by Pfizer, is of primary importance to help making decisions about the necessity of administrating further doses (third only??)

Therefore I think that the ms can be published as it is

We want to thank for your carefully examination of this work. We have incorporated the Reviewers and  editorial recommendations into the text and we also have prepared new tables and figures for better reading. We appreciate very much your assessment of our efforts in preparing this manuscript and considering it well written and timely on an important topic.

Reviewer 2 Report

The study entitled: “Differences in the concentration of anti-SARS-CoV-2 IgG 2 antibodies as a result of post-covid or/and post-vaccination immunization” has a very interesting purpose. The conclusion that pre-vaccine healthy subjects have a greater increase in the antibody titer than convalescents after the second dose is also useful to improve the vaccination campaign.

Q1: I think that it would have been more useful to follow the same patient’s antibody titer over the weeks and then compare it with the others, as you did for eight vaccinated, to reduce sample variability. The number of patients in analysis is high but it would have been a stronger conclusion if the patients had been followed up in their humoral response. Why was the blood sample only performed once per patient? Besides the eight laboratory employees, were other patients followed up?

Q2: There is evidence of a strong variability of the antibody titer, was it justified by comorbidities of patients?

Q3: Three patients had a low antibody titer after the second dose, one of them was 46 years old, were they vaccinated on the same day? Were they affected by any pathologies? 

Q4: With what criteria were selected the 28 patients in which was dosed IFN-gamma?

Q5: The patient who had a re-infection was symptomatic? Were the positivity of the swab confirmed in culture? If so, were the sample tested for variants?

Author Response

Dear Editor and Reviewers,

we are thankful for your evaluation and for your comments.

We were thrilled to learn that it has been conditionally accepted for revision and appreciate the helpful comments and suggestions provided by the reviewer and your editorial team. We have carefully reviewed all the comments and suggestions and revised our manuscript accordingly. Below you will find detailed replies to each reviewer comment and the appropriate reference to the changes made within the manuscript itself. In summary;

  • we have prepared new figures and tables, which in our opinion should be more legible;
  • we have updated the discussion and reference list to reflect the latest publications in the field of post-vaccination serology tests;
  • we changed the description of the methodology of gamma interferon assays, so that it was clear that lymphocyte stimulation was performed on the fresh whole blood sample, and after 24 hours of incubation, we obtained plasma in which we measured the concentration of gamma interferon;
  • we made a linguistic correction of the text.

We believe that the revised version of our manuscript was improved and a detailed response to each of the reviewers' comments will dispel any doubts. We hope that the manuscript is now acceptable for revision and we would like to thank you for your time spent in considering our manuscript.

Sincerely,

Joanna Szczepanek and Andrzej Tretyn,

on behalf of all co-authors

Responses for the Reviewer 2:

The study entitled: “Differences in the concentration of anti-SARS-CoV-2 IgG 2 antibodies as a result of post-covid or/and post-vaccination immunization” has a very interesting purpose. The conclusion that pre-vaccine healthy subjects have a greater increase in the antibody titer than convalescents after the second dose is also useful to improve the vaccination campaign.

Q1: I think that it would have been more useful to follow the same patient’s antibody titer over the weeks and then compare it with the others, as you did for eight vaccinated, to reduce sample variability. The number of patients in analysis is high but it would have been a stronger conclusion if the patients had been followed up in their humoral response. Why was the blood sample only performed once per patient? Besides the eight laboratory employees, were other patients followed up?

In the case of these studies and this research group, the determination was performed once for each patient. Apart from the aforementioned group of laboratory workers, of course. We agree with the reviewer that the multiple determination of the antibody level in a given patient is a very good plan and this idea will be used in the future.

Regular monitoring, especially requiring weekly blood samples, was difficult to implement in a large group. Hence, we have limited ourselves to a narrow but disciplined representative group of people. This constant monitoring was intended to show the general trend in the dynamics of vaccination immunization and serves to verify the trends observed in the general cohort. Being aware of the value of regular measurements for the same people, we included a similar group of people vaccinated with the vector vaccine in such a regular observation program. We have also started a program of carrying out regular determinations of antibody levels (at intervals of two months) in several hundred people. The purpose of these studies, however, is to understand the persistence of the response, not to monitor the dynamics of the response during the first weeks of vaccination.

Bearing in mind how valuable is the information resulting from the constant observation of the same participants, we have updated the figure, taking into account, in our opinion, interesting results obtained this week.

Q2: There is evidence of a strong variability of the antibody titer, was it justified by comorbidities of patients?

There are indeed huge variability of the antibody titer from different people. These differences are probably caused, among others, by the condition of the organism, the efficiency of the immune system or/and the presence of comorbidities. Some of the patients in the research group had a history of chronic diseases. However, we did not have complete medical characteristics of the patients, which made it impossible to perform a complete analysis and correlation of antibody titer for groups of diseases. They were also so diverse that it was impossible to distinguish specific research groups with statistically significant level. This is the purpose of our further experiments, which will allow us to answer the question whether the level of antibody titer differs significantly in the groups of patients suffering mainly from diabetes, hypertension, allergies or asthma.

However, we found some relationships between low antibody levels and the coexistence of diabetes or inflammatory diseases, as well as in people undergoing suppressive therapy. In some people, it was possible to determine the concentration of gamma interferon, on the basis of which we confirmed the presence of stimulated T lymphocytes.

Q3: Three patients had a low antibody titer after the second dose, one of them was 46 years old, were they vaccinated on the same day? Were they affected by any pathologies?

All three people were vaccinated on different days. An 86-year-old woman (78 BAU/ml) and a 46-year-old man (106.02 BAU/ml) did not report comorbidities. The third patient, 80 years old (89 BAU/ml), has Crohn's disease and heart failure.

Q4: With what criteria were selected the 28 patients in which was dosed IFN-gamma?

We had limited possibilities of analysing cellular immunity, therefore we decided to randomly select a diverse group. We wanted these people to include the following representatives: convalescents, people vaccinated without the COVID-19 incident, as well as vaccinated convalescents. We decided that this group should include people from our routine monitoring, people with low levels of anti-SARS-CoV-2 antibodies, people with chronic diseases, and people who fell ill despite vaccination and confirmation of high antibody levels. We realize that such a heterogeneous group, with a small number at the same time, is a weakness of this research. Please remember, however, that the main goal of this study was primarily to analyse the differentiated vaccine response and its comparison in the context of SARS-CoV-2 infection (or its absence) and to show the dynamics of the humoral response in the first weeks of vaccination. The analysis of cellular immunity complements the main research.

Q5: The patient who had a re-infection was symptomatic? Were the positivity of the swab confirmed in culture? If so, were the sample tested for variants?

In our study, 3 vaccinated subjects tested positive for PCR. These people (employee of the sanitary and epidemiological station, employee and inmate of a nursing home) are routinely tested for these institutions, so they obtained the result after screening. The infections were asymptomatic in two of them (including the vaccinated convalescent who was re-infected). The nursing home patient required later hospitalization, which was probably a consequence of old age and the coexistence of chronic diseases. In Poland, variant determinations are not performed in standard diagnostics, therefore we do not have information about variants with which the vaccinated persons were infected.

Reviewer 3 Report

  • According to the journal’s webpage:

Cells covers every topic related to cell biology and physiology, molecular biology, and biophysics. Thus, our major focus is on experimental cytology rather than on clinical and epidemiological studies.

The submission by Tretyn and coworkers, on SARS-CoV-2 immune responses studied by clinical (ex-vivo) samples, in scope not at all matches this journal.

  • While the clinical samples are numerous (n, 477), their division into 16 subgroups makes the material extremely heterogenous and difficult for the reader to follow; without any obvious (scientific or practical) gain.

  • According to Methods, a single antibody assay was used – a standard commercial (EUROIMMUNE’s) IgG ELISA, which according to PubMed has been used previously in >100 scientific articles. I.e., the current submission lacks methodological novelty.

  • At places the authors address their “neutralizing antibody” results; without any evidence of such an assay (which are entirely different from the ELISA above) in reality having been used.

  • Whereas 2.3. “Cellular response analysis – cell stimulation” wishes to give the impression of SARS-CoV-2-specific T-cell studies having been performed with a newly launched commercial kit, the reader remains fully ignorant of which cell type(s) the present results actually were derived from; and based on what? Even worse, on lines 149-50 it reads “If activation-capable cells are present in plasma, they are stimulated to secrete interferon...” In reality, human plasma is cell-free, i.e. contains no lymphocytes.

  • In results this piece of work is merely confirmatory to existing literature. Its main characteristic, as mentioned, is clinical heterogeneity.

  • In writing this submission is substandard, both scientifically and linguistically.

Author Response

Dear Editor and Reviewers,

we are thankful for your evaluation and for your comments.

We were thrilled to learn that it has been conditionally accepted for revision and appreciate the helpful comments and suggestions provided by the reviewer and your editorial team. We have carefully reviewed all the comments and suggestions and revised our manuscript accordingly. Below you will find detailed replies to each reviewer comment and the appropriate reference to the changes made within the manuscript itself. In summary;

  • we have prepared new figures and tables, which in our opinion should be more legible;
  • we have updated the discussion and reference list to reflect the latest publications in the field of post-vaccination serology tests;
  • we changed the description of the methodology of gamma interferon assays, so that it was clear that lymphocyte stimulation was performed on the fresh whole blood sample, and after 24 hours of incubation, we obtained plasma in which we measured the concentration of gamma interferon;
  • we made a linguistic correction of the text.

We believe that the revised version of our manuscript was improved and a detailed response to each of the reviewers' comments will dispel any doubts. We hope that the manuscript is now acceptable for revision and we would like to thank you for your time spent in considering our manuscript.

Sincerely,

Joanna Szczepanek and Andrzej Tretyn,

on behalf of all co-authors

Responses for the Reviewer 3:

According to the journal’s webpage:

Cells covers every topic related to cell biology and physiology, molecular biology, and biophysics. Thus, our major focus is on experimental cytology rather than on clinical and epidemiological studies.

The submission by Tretyn and coworkers, on SARS-CoV-2 immune responses studied by clinical (ex-vivo) samples, in scope not at all matches this journal.

On April 19, I contacted the Cells editorial office (Yuma Zhang Assistant Editor, MDPI AG) regarding our manuscript proposal (it was my reaction to the invitation to submit the manuscript to Cells). Moreover based on the abstract sent, the editor considered it appropriate for the journal, and it was the editor's suggestion to submit a submission to the special issue “Cellular Immunology and COVID-19”

After submission our manuscript was read by the journal's editors. Since it was sent for review, it had to be qualified as matching the subject of the journal. Looking at the title of the special issue, it is difficult to say that our article deals with a different topic.

While the clinical samples are numerous (n, 477), their division into 16 subgroups makes the material extremely heterogenous and difficult for the reader to follow; without any obvious (scientific or practical) gain.

The number of subgroups is based on the division into 3 main groups: unvaccinated, vaccinated without prior SARS-CoV-2 infection and vaccinated convalescents. This is a standard division that we find in studies from the literature on the subject. The next division consists of periods: that is, before the vaccination, after taking the first dose and after taking the second dose. This is also the standard approach found in vaccine response manuscripts. A unique feature of our research is showing the dynamics of changes week after week. Reducing the number of groups would make our research similar to what is shown in other articles. It would also make it impossible to show the rate of increase in the concentration of antibodies up to about 2-3 weeks after taking the second dose, and then the rate of decrease of this concentration in the following weeks. Hence, we would not like to reduce the number of groups, but we have changed the way of data visualization. Thus, we hope that the results have become more readable and clear.

According to Methods, a single antibody assay was used – a standard commercial (EUROIMMUNE’s) IgG ELISA, which according to PubMed has been used previously in >100 scientific articles. I.e., the current submission lacks methodological novelty.

We are aware that the time of a global pandemic is a time of intensive scientific research, therefore it is not surprising that studies in the field of COVID-19 are appearing rapidly. We are aware that we are participating in a kind of race for the priority of publication. The editorial time of a given manuscript is not without significance. Please note that in Poland, the first vaccination was carried out only on December 27, 2020. It was only in January that vaccination on a larger scale was started. The studies required planning, organizing the equipment and recruiting participants (minimum 2 weeks after the first dose to the 5th week after the second dose), which took us until the end of February. The following weeks are devoted to data processing, analysis and preparation of the manuscript. On April 2, we submitted the manuscript to ResearchSquare, where it was published on April 5. Most of the websites where it is available as a preprint has relatively good statistics. As I wrote on April 19, we consulted the Cells editorial staff on the interest in the manuscript, and after 5 weeks (on May 27) we received a complete reviews and the editor's decision. We have no influence on the editorial cycle, but we find it difficult to agree with the lack of influence on the knowledge of the vaccine response.

As I wrote above, a unique feature of our research is to show the dynamics of changes week after week (and to our knowledge there are no such studies). In addition, the studies covered a relatively large group of participants compared to other studies. It shows not only the manifestation of the humoral response but also the elements of the cellular response. We follow the literature on an ongoing basis and made additions to the discussion on the latest results presented by other research teams.

It is difficult to talk about an innovative method of determining the level of antibodies here. ELISA is the most widely used method for such analyzes. Moreover, it is specific and cheap compared to other methods. We wanted to focus on examining as many patients as possible in order to obtain the broadest possible picture of the result. When it comes to using a specific set from Euroimmun, it was certainly innovative in the course of the experiments, and due to the specification, in our opinion, the most appropriate.

At places the authors address their “neutralizing antibody” results; without any evidence of such an assay (which are entirely different from the ELISA above) in reality having been used.

Thank you for paying attention to the naming details. In the study, we analysed IgG anti-SARS-CoV-2 antibodies using the Elisa method, we did not perform neutralization tests. We consciously chose one set for serological analyses, which seemed to us the most suitable for large-scale studies, because of its advantages. According to the manufacturer's characteristics, the advantages of the Anti-SARS-CoV-2 QuantiVac ELISA (IgG) test include the test of IgG antibodies with neutralizing properties directed against the S protein (spike) of the SARS-CoV-2 virus, excellent correlation of the test with the new WHO reference material (NIBSC code: 20/136) and 100% compliance with the PRNT50 neutralization test (which is the "gold standard" for measuring the effectiveness of antibodies in neutralizing viruses). In order to avoid confusion, we have corrected the nomenclature and terms of neutralizing antibodies, changed to antibodies with neutralizing properties or anti-SARS-CoV-2 antibodies in the IgG class

Whereas 2.3. “Cellular response analysis – cell stimulation” wishes to give the impression of SARS-CoV-2-specific T-cell studies having been performed with a newly launched commercial kit, the reader remains fully ignorant of which cell type(s) the present results actually were derived from; and based on what? Even worse, on lines 149-50 it reads “If activation-capable cells are present in plasma, they are stimulated to secrete interferon...” In reality, human plasma is cell-free, i.e. contains no lymphocytes.

Thank you for this attention. Revised version of the manuscript has been supplemented with information on the operating principle of the kit used in the experiment. Moreover, in the manuscript at line 149-150, there was an error that has been corrected. T cells in whole blood were stimulated to synthesize and release gamma interferon, the level of which was then determined in plasma obtained from whole blood after stimulation. We sincerely apologize for this error.

In results this piece of work is merely confirmatory to existing literature. Its main characteristic, as mentioned, is clinical heterogeneity.

As I wrote above, our work has its own unique features (large numbers and detailed analysis week by week). To our knowledge, none of the published works presents several months of observations of changes in the level of antibodies performed in the same participants. It was published in the preprint on April 5, when there were not many studies on the analysis of the post-vaccination level of antibodies to SARS-CoV-2, so it is not our work that confirms the existing literature (please remember the importance of the editorial cycle of the publishing house). Besides, I think that the confirmation of specific observations by independent teams is a big advantage and strengthening of importance.

In writing this submission is substandard, both scientifically and linguistically.

The manuscript has been thoroughly verified in terms of any editing and linguistic errors. We have also made a professional English edition (by MDPI English editing service) .

Reviewer 4 Report

This article could be very interesting and useful to understand the efficacy of humoral response in population immunized to SARS-CoV-2 as result of coronavirus infection and/or vaccination by mRNA technology vaccines.

However, this paper is not suitable because it is seriously inaccurate and confusing. It is not at all clear how IFN-gamma can be assayed in plasma since plasma does not contain cells. Are Th cells and Th1 lymphocytes isolated? Where is this shown? Did the authors perhaps co-stimulate Th1 with plasma (isolated how?). Therefore, this cytokine may be produced by other cellular elements present in the whole blood of the immune system. What happened to the data on CD8+ lymphocytes?

Moreover the subdivision in so many subgroups (may be too many) complicates everything and for this to make a correlation the authors use the Spearman index. Figure 2 that is not very clear should be improved also to understand if they found a difference of IgG between the two sexes. In both cases, they should provide a working hypothesis.

Author Response

Dear Editor and Reviewers,

we are thankful for your evaluation and for your comments.

We were thrilled to learn that it has been conditionally accepted for revision and appreciate the helpful comments and suggestions provided by the reviewer and your editorial team. We have carefully reviewed all the comments and suggestions and revised our manuscript accordingly. Below you will find detailed replies to each reviewer comment and the appropriate reference to the changes made within the manuscript itself. In summary;

  • we have prepared new figures and tables, which in our opinion should be more legible;
  • we have updated the discussion and reference list to reflect the latest publications in the field of post-vaccination serology tests;
  • we changed the description of the methodology of gamma interferon assays, so that it was clear that lymphocyte stimulation was performed on the fresh whole blood sample, and after 24 hours of incubation, we obtained plasma in which we measured the concentration of gamma interferon;
  • we made a linguistic correction of the text.

We believe that the revised version of our manuscript was improved and a detailed response to each of the reviewers' comments will dispel any doubts. We hope that the manuscript is now acceptable for revision and we would like to thank you for your time spent in considering our manuscript.

Sincerely,

Joanna Szczepanek and Andrzej Tretyn,

on behalf of all co-authors

Responses for the Reviewer 4:

This article could be very interesting and useful to understand the efficacy of humoral response in population immunized to SARS-CoV-2 as result of coronavirus infection and/or vaccination by mRNA technology vaccines.

However, this paper is not suitable because it is seriously inaccurate and confusing. It is not at all clear how IFN-gamma can be assayed in plasma since plasma does not contain cells. Are Th cells and Th1 lymphocytes isolated? Where is this shown? Did the authors perhaps co-stimulate Th1 with plasma (isolated how?). Therefore, this cytokine may be produced by other cellular elements present in the whole blood of the immune system. What happened to the data on CD8+ lymphocytes?

Thank you very much for noting this method description error. Interferon-gamma release assay (IGRA) was performed for quantitative determination of the IFN-γ release of SARS-CoV-2-specific T cells. T-cell stimulation were performed in heparinised whole blood (not plasma!!) based on the highly immunogenic spike protein and subsequent determination of released IFN-γ (in plasma) by means of ELISA. In our study, we used the SARS-CoV-2-IGRA assay which is helpful in determining large scale SARS-CoV-2 specific T cell responses with high accuracy. Through specific antigen (and control mitogen) stimulation, we were able to analyze T cell responses against the SARS-CoV-2 structural protein using a simple, fast and efficient approach. In the peripheral blood using automatic, easy to use the interferon gamma release test, we could indirectly infer the reactivity of T lymphocytes and correlate them with serological data. However, we were unable to directly analyze the lymphocyte subpopulation. It is a methodological approach quite often used in the analysis of the cellular response specific for SARS-CoV-2 (e.g. Murugesan et al. 2020, Brand et al. 2021).

Information on the operating principle of the IGRA test has been supplemented in subsection 2.3. Moreover, an error has crept into the Materials and Methods of the manuscript. We have carried out all stages in accordance with the manufacturer's recommendations and there is no methodological error in them. It is a mistake to have a lapse in the text that we apologize for.

We recognize that a thorough understanding of the mechanisms of immune memory against SARS-CoV-2 requires the evaluation of its various components, including B cells, CD8 + (cytotoxic) T cells and CD4 + (helper) T cells. So far it has been proven that as a result of infection with SARS-CoV-2 virus, T lymphocytes are strongly stimulated. It has also been proven that the S protein of the SARS-CoV-2 virus is the antigen activating these lymphocytes. Initial observations of several months of convalescents show that almost 100% of patients after COVID-19 suffer from SARS-CoV-2 specific CD4 + T lymphocytes, and in 70% CD8 + T lymphocytes. For vaccinated persons, such data is still incomplete. Please also note that the analysis of the cellular response (among others because we had no way of differentiating the lymphocyte subpopulations) complements the mainstream research of monitoring the humoral response in the first weeks after vaccination.

Moreover the subdivision in so many subgroups (may be too many) complicates everything and for this to make a correlation the authors use the Spearman index. Figure 2 that is not very clear should be improved also to understand if they found a difference of IgG between the two sexes. In both cases, they should provide a working hypothesis.

The number of subgroups is based on the division into 3 main groups: unvaccinated, vaccinated without prior SARS-CoV-2 infection and vaccinated convalescents. This is a standard division that we find in studies from the literature on the subject. The next division consists of periods: that is, before the vaccination, after taking the first dose and after taking the second dose. This is also the standard approach found in vaccine response manuscripts. A unique feature of our research is showing the dynamics of changes week after week. Reducing the number of groups would make our research similar to what is shown in other articles. It would also make it impossible to show the rate of increase in the concentration of antibodies up to about 2-3 weeks after taking the second dose, and then the rate of decrease of this concentration in the following weeks. Hence, we would not like to reduce the number of groups, but we have changed the way of data visualization. Thus, we hope that the results have become more readable.

As we wrote in the manuscript, no correlation was found between the level of antibodies and the age of patients or gender. We presented the results as an illustration, because although we can see relatively higher levels of antibodies in women, the differences were not statistically significant.

Round 2

Reviewer 4 Report

The manuscript has been almost improved, but the authors should take into account another usefull and similar published paper that has not been mentioned: Petrone L, Petruccioli E, Vanini V, Cuzzi G, Najafi Fard S, Alonzi T, Castilletti C, Palmieri F, Gualano G, Vittozzi P, Nicastri E, Lepore L, Antinori A, Vergori A, Caccamo N, Cantini F, Girardi E, Ippolito G, Grifoni A, Goletti D. A whole blood test to measure SARS-CoV-2-specific response in COVID-19 patients. Clin Microbiol Infect. 2021 Feb;27(2):286.e7-286.e13. doi: 10.1016/j.cmi.2020.09.051.

Minor revision

Lines 65-67 Specific CD4+ T cells differentiate into Th1 and Tfh cells: the former produce IFNγ and cytokines, and the latter, in addition to activating B cells, Tfh cells are crucial for the proper functioning of the neutralizing antibodies

Please delete "and cytokines" because they do not indicate which ones and if they are not pertinent it does not make sense

Line 178. Please specify the mitogen that represents the positive control.

Lines 179-181 Please remove this sentence “The plasma obtained from this tube is was used to verify whether the sample contains contained a sufficient number of cells and whether they have had a sufficient ability to become activate”

Page 11 please replace the incorrect abbreviation for interferon gamma INFϒ with IFNϒ, and use the abbreviation 

Line 324 insert mean leavel of…

Line 330 replace “a marker of lymphocyte activity” with IFNϒ

 Line 331 replace “marked”  with was “detected in”

Line 338 “in another our study conducted by us”

Author Response

We would like to thank the Reviewer for their insightful comments and suggestions. We have responded to all questions (Italics) and have made modifications in the manuscript (highlighted) to address all concerns and suggestions raised by the Reviewer. Below is a detailed responses.

Sincerely,

Joanna Szczepanek

on behalf of all co-authors

The manuscript has been almost improved, but the authors should take into account another usefull and similar published paper that has not been mentioned: Petrone L, Petruccioli E, Vanini V, Cuzzi G, Najafi Fard S, Alonzi T, Castilletti C, Palmieri F, Gualano G, Vittozzi P, Nicastri E, Lepore L, Antinori A, Vergori A, Caccamo N, Cantini F, Girardi E, Ippolito G, Grifoni A, Goletti D. A whole blood test to measure SARS-CoV-2-specific response in COVID-19 patients. Clin Microbiol Infect. 2021 Feb;27(2):286.e7-286.e13. doi: 10.1016/j.cmi.2020.09.051.

We thank the reviewer for the constructive comment. As suggested, we have included a reference to the publication in the discussion. Thus, we also changed further citations. Please see page 10, lines 321-324.

Minor revision

Lines 65-67 Specific CD4+ T cells differentiate into Th1 and Tfh cells: the former produce IFNγ and cytokines, and the latter, in addition to activating B cells, Tfh cells are crucial for the proper functioning of the neutralizing antibodies

Please delete "and cytokines" because they do not indicate which ones and if they are not pertinent it does not make sense

We have removed from the sentence "and cytokines" as suggested.

Line 178. Please specify the mitogen that represents the positive control.

In our study, we used commercial EUROIMMUN kit (SARS-CoV-2 Interferon Gamma Release Assay (IGRA)) and therefore we do not know the specifics of the mitogen with which the STIM tubes were coated. We asked our distributor with this inquiry, but unfortunately we did not receive detailed information due to the manufacturer's patent. We only know that it is a test tube coated with a mitogen that causes non-specific secretion of interferon-gamma. Thus, the obtained plasma serves to verify that the sample contains a sufficient number of cells and that they have sufficient capacity activation.

Lines 179-181 Please remove this sentence “The plasma obtained from this tube is was used to verify whether the sample contains contained a sufficient number of cells and whether they have had a sufficient ability to become activate”

As mentioned above, we don't know exactly what mitogen was used by the manufacturer of the kit. Hence, we applied the principle that, in order to precisely describe the methodology, we will characterize each of the test tubes used for the stimulation of whole blood. We wrote analogous descriptions for BLANK, STIM and TUBE tubes. By removing the aforementioned sentence, we would disturb the integrity of the description of this part of the methodology. Therefore, we would rather leave this sentence for people unfamiliar with the IGRA test methodology.

Page 11 please replace the incorrect abbreviation for interferon gamma INFϒ with IFNϒ, and use the abbreviation

Thank you for Your vigilance - we have corrected this mistake of course.

Line 324 insert mean leavel of…

We have completed this wording as suggested. Thank you for Your comment.

Line 330 replace “a marker of lymphocyte activity” with IFNϒ

We have made this correction in the text.

 Line 331 replace “marked”  with was “detected in”

Done.

Line 338 “in another our study conducted by us”

Thank you for your suggestion. We corrected this sentence.
